# The English Debating Self-Efficacy Scale: Scale development, validation, and psychometric properties

**Fanghua Liu[1], Yanchao Yang[2]\*, Feng (Robin) Wang[3], Wangze Li[4]**

**1** School of Graduates, Changchun Normal University, Changchun, Jilin Province, People's Republic of China, **2** Institute of International Language Services Studies, Macau Millennium College, Macau SAR, People's Republic of China, **3** School of Translation Studies, Shandong University at Weihai, Weihai, Shandong Province, People's Republic of China, **4** College of Materials Science and Engineering, North China University of Science and Technology, Tangshan, Hebei Province, People's Republic of China

\* 124985392@qq.com

**Data Availability Statement:** All relevant data are within the manuscript and its Supporting Information files.

**Funding:** The author(s) received no specific funding for this work.

## Abstract

The importance of English debate in fostering critical thinking and the role of self-efficacy in enhancing confidence and performance in this domain are widely acknowledged. However, a significant gap exists in the literature regarding the measurement of self-efficacy specifically within English debate. This research seeks to fill this gap by developing and validating an English Debate Self-Efficacy Scale (EDSS). Using a sample of 1,259 participants from an independent college in Hebei Province, China, the study divided participants into two groups: 613 for exploratory factor analysis (EFA) and 646 for confirmatory factor analysis (CFA), with convenience sampling as the chosen methodology. EFA revealed three core dimensions of debate-related self-efficacy: Language proficiency (Cronbach's Alpha = .894), Debating skills (Cronbach's Alpha = .861), and Team collaboration (Cronbach's Alpha = .831). Subsequent CFA validation with an independent sample confirmed the scale's structure, demonstrating strong structural, convergent, and discriminant validity. Additionally, significant correlations between the English Debate Self-Efficacy Scale and the English Proficiency Self-Efficacy Scale supported the scale's criterion validity. These findings underscore the scale's potential as a reliable tool for assessing self-efficacy in English debate contexts, offering valuable insights for research, teaching, and training in educational settings. Limitations related to sample representativeness and research design were also discussed, providing a foundation for future studies to expand upon. In conclusion, the English Debate Self-Efficacy Scale (EDSS) is a reliable and valid instrument for measuring self-efficacy in the context of English debate.

## 1. Introduction

### 1.1 Research background

Debate is an activity where opposing sides argue about a predetermined topic. The team that presents the most relevant, strong, and evidence-based arguments is declared the winner [1].

**Competing interests:** The authors have declared that no competing interests exist.

Debaters are tasked with organizing and articulating their viewpoints within a limited time-frame, while engaging in compelling arguments with their opponents.

Debates serve as a powerful platform for enhancing civic education and civic engagement among university students [2–6]. By actively participating in debates, students can significantly strengthen their public awareness and sense of social responsibility. The debate format encourages deep engagement with pressing social and political issues, cultivating a more nuanced understanding of complex public policies and their real-world implications. This, in turn, inspires a heightened sense of civic duty and agency, as students witness the impact their ideas and advocacy can have. Through the debate experience, university students are motivated to continue actively shaping their communities and society at large upon graduation.

Additionally, debates emerge as a transformative pedagogical approach for cultivating critical thinking skills in university students [7–9]. The very structure of debates requires rigorous analytical reasoning, pushing students to gather, evaluate, and synthesize information from diverse, credible sources. This development of information literacy lays the groundwork for more nuanced, evidence-based decision-making. Moreover, the imperative to construct coherent, persuasive arguments forces students to deconstruct issues from multiple angles. In building their cases, they must anticipate counterarguments and devise compelling rebuttals. This process of structured, logical reasoning challenges students to move beyond superficial understanding and engage in deeper, more critical analysis.

In addition, participation in debate competitions provides university students with unparalleled opportunities to develop and refine their communication skills [10–12]. At the heart of the debate process lies the imperative to engage in dynamic, responsive exchange—a central tenet of effective communication. the collaborative nature of debate teams requires students to hone their interpersonal communication abilities. In working with teammates to craft their overall strategy, students learn to negotiate, compromise, and synergize their individual strengths. This teamwork dynamic mirrors the give-and-take of communication in real-world settings, equipping students with invaluable interpersonal skills.

Furthermore, in modern academia, there is a growing focus on the importance of English debate in college settings [8, 13–16]. As a result, Chinese institutions are increasingly supporting English debate clubs and related activities. These efforts aim to establish English debate as a valuable tool for enhancing English language proficiency among college students [17]. Using debates in English as a Foreign Language (EFL) classes provides students with a practical opportunity to apply their English language skills in real-life situations [18]. For instance [19], conducted a qualitative study to examine the advantages of participating in a debating club for improving students' speaking skills. The results showed that active engagement in debates led to improvements in five specific speaking skill traits and three additional soft skill dimensions. The improvements included increased fluency, expanded vocabulary acquisition, better grasp of debate concepts, improved pronunciation accuracy, enhanced grammatical proficiency, strengthened critical thinking abilities, enhanced collaborative learning, and improved problem-solving skills.

## 1.2 Statement of research problem

Given the impact of debate on students, particularly the role of English-language debates in enhancing language skills, it becomes essential to understand their self-perceptions regarding their abilities to excel in this specific domain. Self-efficacy, defined as an individual's belief in their capacity to execute tasks successfully, is a well-documented predictor of actual performance outcomes [20]. In the competitive arena of English debate, a student's self-efficacy—that is, their confidence in their ability to effectively participate and succeed in English debate—emerges as a crucial determinant of their actual debate skills and results. Notwithstanding

the significance of this factor, there is a notable absence of a validated tool specifically designed to measure English debate self-efficacy among university students.

### 1.3 Research purpose

Consequently, this study aims to develop and validate an English Debate Self-Efficacy Scale, a novel instrument designed to accurately evaluate students' confidence in their debate-related skills. By illuminating the context-, domain- and task-specific aspects of self-efficacy within English debate, this groundbreaking research has the potential to guide the creation of tailored interventions that enhance students' English communication skills and foster their development as proactive global citizens and leaders.

### 1.4 Research significance

The significance of this research lies in its potential to fill a crucial gap in the existing literature on self-efficacy in English debate contexts. By developing the English Debate Self-Efficacy Scale (EDSS), this study can contribute to the theoretical understanding of self-efficacy and provide a practical tool for educators and researchers. The insights gained from this research may inform the design of targeted programs and strategies that can boost students' confidence and skills in debate. Furthermore, by emphasizing the importance of self-efficacy in enhancing communication skills, this study can support the broader goal of preparing students to become proactive global citizens and leaders, equipped to engage effectively in diverse contexts. Ultimately, the EDSS can serve as a valuable resource for advancing both academic inquiry and practical applications in educational settings.

## 2. Literature review

This section explored the concept of self-efficacy, providing a brief overview of the sources that could strengthen it. It also highlighted common research focuses across various fields. Subsequently, it examined self-efficacy's nature as domain-, context-, and task-specific. Lastly, it addressed the notable scarcity of appropriate scales designed specifically to measure self-efficacy in the context of English debate.

### 2.1 Self-efficacy

Bandura defines self-efficacy as "people's judgment of their capabilities to organize and execute courses of action required to attain designated types of performances" in the future [20]. According to [20, 21], self-efficacy beliefs are formed through various sources, including enactive attainment or mastery experience, vicarious experience, verbal persuasion, and physiological state.

Over the past decades, learners' perceived self-efficacy and beliefs regarding English language learning have emerged as significant issues in education. One study [22] conduct a comprehensive review and thematic synthesis of the growing empirical research conducted on EFL self-efficacy in the past decade and provided an overview of the main research topics, including the relationship with language-learning anxiety [e.g., 23–29], motivation [e.g., 27, 29–33], self-esteem [e.g., 34–38], value [e.g., 39–42], goals [e.g., 43, 44], teacher-student relationships [e.g., 45–48], classroom setting [e.g., 49, 50] and language achievement [e.g., 25, 27, 29, 30, 51–56].

### 2.2 Scarcity of proper scale for English Debating Self-Efficacy

Although there has been considerable research on self-efficacy in the context of English as a Foreign Language (EFL), covering a wide range of topics, there is still a lack of literature specifically focusing on measuring self-efficacy related to specific English tasks.

It is important to emphasize that self-efficacy is typically considered context-specific, domain-specific, and task-specific [20, 57]. This implies that the measurement of self-efficacy requires customized scales that are suitable for different contexts, domains, and tasks. Self-efficacy beliefs can vary based on the specific situation, domain, or task. Individuals may exhibit varying levels of confidence and perceived competence in different contexts or areas of their lives. [21] argues that the strength of self-efficacy as a predictor variable comes from its specificity, and with such a great difference in the cognitive demands placed on learners by specific skill or task, it is necessary to develop measures of self-efficacy that are specific to certain skills. Therefore, given self-efficacy needs to be measured specifically rather than generally [58–60]. Accurate assessment of self-efficacy requires context-specific scales tailored to the particular domain or task. These scales consider the unique challenges and characteristics of the specific context, enabling researchers to capture the intricacies of self-efficacy within different domains and tasks. This approach provides a more precise representation of individuals' self-efficacy beliefs in those specific areas.

Although, effective self-efficacy measures have been developed in foreign language education research, such as speaking [61, 62], listening [24, 63–67], reading [68–72] and writing [68, 73–80], English debate self-efficacy is relatively under-researched. Though English debate, English public speaking and English oral communication fall under the umbrella of sopken English, their fundamental natures differ significantly. Firsr, debate is an interactive and competitive format that demands not only linguistic proficiency but also critical thinking, quick reasoning, strategic interaction, and effective team collaboration. These components are essential as they collectively contribute to a debater's ability to perform successfully in a dynamic environment where arguments must be constructed, deconstructed, and defended on the fly. Second, public speaking generally involves delivering information or persuasion in a more monologic manner, focusing primarily on effective delivery and audience engagement. Third, English oral communication is generally informal and spontaneous, and do not follow a structured format. The primary purpose is often social interaction, information exchange, or casual discussion on various topics. While scales for measuring English oral communication and public speaking skills can provide valuable insights into general linguistic abilities and the confidence to speak in front of an audience, they cannot be directly applied to the context of English-language debating. This limitation arises because self-efficacy is highly specific to the context in which the skills are applied, the domain of knowledge required, and the specific tasks involved.

Therefore, this study aims to develop and validate a scale specifically designed to measure English debate self-efficacy. By addressing this research gap, we hope to provide a valuable tool that can enhance our understanding of the unique skills and confidence required for effective debating, ultimately contributing to improved educational practices and outcomes in this important area of language use.

## 3. Methods

### 3.1 Research design

This study employs a quantitative research design aimed at developing and validating the English Debate Self-efficacy Scale (EDSS). The research follows a two-phase approach, incorporating Exploratory Factor Analysis (EFA) to identify the underlying dimensions of self-efficacy in English debate, followed by Confirmatory Factor Analysis (CFA) to validate the scale's structure and ensure its reliability and validity. Specifically, reliability is assessed through internal consistency, measured using Cronbach's Alpha for each dimension of the scale. For validity, structural validity, convergent validity, discriminant validity and criterion-related validity were assessed.

## 3.2 Participant description

This research was conducted at a university located in northern China, where participants were selected using convenience sampling to ensure accessibility and ease of participation. The choice of this sampling method is justified by the fact that a co-author is employed at the university, which facilitates access to a sufficient number of respondents to complete the questionnaires. Additionally, this connection helps to create an environment conducive to participation, as students are more likely to engage in research conducted within their familiar academic setting. All participants involved in the study were university students. Data was gathered using the Wenjuanxing platform, a widely used survey tool in China known for its user-friendly interface and efficient data management capabilities. The Wenjuanxing platform was specifically chosen for its ability to facilitate easy access and streamline the data collection process, allowing participants to respond at their convenience. The data collection occurred over a one-week period, from September 10 to 17, 2023. During this time, participants were invited to complete the questionnaire, which was designed to take approximately five minutes. This brief duration aimed to maximize participation while minimizing participant fatigue. After data collection, the Mahalanobis distance method was applied to identify and filter out careless or inconsistent responses, ensuring the reliability of the dataset. This rigorous cleaning process resulted in a final sample of 1,259 participants, who were deemed suitable for further analysis. This careful selection process underscores the integrity and validity of the findings derived from this research.

The sample was randomly split into two groups using SPSS. The first group, consisting of 613 participants, was utilized for Exploratory Factor Analysis (EFA), while the second group, with 646 participants, was assigned to Confirmatory Factor Analysis (CFA). Table 1 provides the detailed demographic information.

The sample sizes for both EFA and CFA were deemed sufficient, following [81] guideline, which advises a minimum of ten respondents per item. This method ensures a proper participant distribution, facilitating a comprehensive analysis of the factors being studied. By adhering to this standard, the study can reliably assess the underlying constructs and draw valid conclusions from the data.

## 3.3 Instruments

**3.3.1 English Debate Self-Efficacy Scale (EDSS).**   To explore the concept of English debate self-efficacy, we conducted interviews with four students who had prior experience in English debate competitions. These participants were selected for their diverse backgrounds and varying levels of success in debates, allowing us to gather a wide range of perspectives. During the interviews, we aimed to uncover the specific experiences in competitive settings that represented their confidence and perceived skills in debating. By prompting participants to reflect on these key experiences, we sought to identify the elements that embody their self-

**Table 1. Demographic information.**

| Demographic Variable | Category | EFA Dataset | | CFA Dataset | |
|---|---|---|---|---|---|
| | | Frequency | Percent | Frequency | Percent |
| Gender | Male | 125 | 20.40% | 138 | 21.40% |
| | Female | 488 | 79.60% | 508 | 78.60% |
| Birthplace | Urban | 319 | 52.00% | 302 | 46.70% |
| | Rural | 294 | 48.00% | 344 | 53.30% |
| Total | | 613 | 100.00 | 646 | 100.00 |

efficacy in English debate, providing valuable insights for the development of a robust self-efficacy scale.

After conducting the interviews, we collected and organized the data to facilitate thorough analysis. Utilizing MAXQDA software, we meticulously analyzed the interview transcripts through a process of systematic coding. Initially, we coded the transcripts to identify recurring ideas and notable statements related to the participants' experiences. As we examined these codes, we began to group related concepts together, leading to the emergence of three primary dimensions: debating skills, collaboration, and English language proficiency. Each dimension was derived from the participants' reflections on their capabilities and interactions during competitive settings, highlighting the essential elements of self-efficacy in English debate. Altogether, 16 items were generated.

Subsequently, we sought feedback from two experienced teachers who had coached English debate teams for several years. Their expertise provided valuable insights into the relevance and clarity of the 16 items and dimensions of the questionnaire. We presented them with the initial set of items and asked for their assessments regarding the appropriateness and effectiveness of each item in measuring self-efficacy in English debate. After thorough discussions, they recommended removing three items that were either redundant or unclear in their wording or not representative. As a result of their constructive feedback, we refined the questionnaire, leading to a final set of 13 items that effectively capture the key dimensions of self-efficacy for further analysis. These items were categorized into three factors as shown in S1 Table: debating skills, consisting of 4 items, collaboration, consisting of 3 items, and English language proficiency, consisting of 6 items.

Based on the refined item set, the operational definition for English Debate Self-Efficacy in this study refers to self-reported confidence of university students in their abilities to successfully participate in and contribute to English-language debates. It is framed around the three proposed distinct but interrelated dimensions: Debating Skills (encompassing the mastery of formal debating techniques), Team collaboration (assessing the effectiveness of teamwork and interpersonal interactions within a debate setting) and Language Proficiency (evaluating the fundamental language skills essential for engaging in English-language debates).

**3.3.2 English Public Speaking Self-Efficacy Scale (EPSSS).** English Public Speaking Self-Efficacy Scale was originally developed and validated [82] using a sample of 406 English as a Foreign Language (EFL) learners and, since then, it has been widely adopted and utilized in various empirical studies [e.g. 83–88]. This scale is designed to measure individuals' self-efficacy beliefs in English public speaking and consists of 12 items. Respondents rate each item on a 5-point scale, ranging from 1 (strongly disagree) to 5 (strongly agree). The scale assesses EPS self-efficacy across four dimensions, namely, language competence (Cronbach's Alpha = .69), organization competence (Cronbach's Alpha = .74), topic competence (Cronbach's Alpha = .77), and delivery competence (Cronbach's Alpha = .72).

## 3.4 Analytical procedure

The study first conducted item-level analysis, which included descriptive statistics and item discrimination, to evaluate the variability and discriminative power of each item. Next, an Exploratory Factor Analysis (EFA) was performed to identify the factorial structure of the 13-item scale. Following this, a Confirmatory Factor Analysis (CFA) was conducted to assess and refine the model's fit. The suitability of the EFA was first evaluated using Bartlett's test of sphericity and the Kaiser-Meyer-Olkin (KMO) measure. Subsequently, univariate and multivariate normality were examined before conducting the CFA. To assess construct validity, several criteria were employed, including the chi-square to degrees of freedom ratio, root mean

square error of approximation (RMSEA), standardized root mean square residual (SRMR), Comparative Fit Index (CFI), and Tucker-Lewis Index (TLI). Convergent validity was then examined using factor loadings, composite reliability (CR), and average variance extracted (AVE). Discriminant validity was assessed using the Heterotrait-Monotrait Ratio of Correlations (HTMT). Finally, criterion-related validity was determined by analyzing the correlations with participants' responses to EPSSS.

### 3.5 Ethical considerations

To maintain research integrity, we adopted various ethical measures. Initially, approval was obtained from the Research Ethics Review Board at the College of Materials Science and Engineering, North China University of Science and Technology (Ref. No.: HLCL2022080502) to ensure our study adhered to ethical standards. Authorization for data collection was also granted by relevant school authorities, legitimizing our research activities in the designated region.

In terms of informed consent, we took multiple precautions. The digital consent form was integrated into the electronic questionnaire, and students were required to confirm their participation by clicking "agree to participate" before proceeding. The form clearly communicated their rights, emphasizing that they could withdraw from the study at any point without any negative consequences. Additionally, to protect student privacy, no identifiable information such as student IDs or names was collected. The data was securely stored on a password-protected computer, accessible only to the research team.

## 4. Results

### 4.1 Descriptive analysis

The findings of the item-level analyses indicated that each item had a normal distribution, as evidenced by the skewness and kurtosis values being less than ±1.

### 4.2 Item-total correlation

Additionally, this study utilized item-total correlations to assess the relationship between each item and the overall score of the scale. The findings indicated a moderate positive correlation, ranging from 0.446 to 0.645, between each item and the total score. This suggests that the items are effective in measuring the underlying variable being evaluated.

### 4.3 Item discrimination

Participants were categorized into high-scoring and low-scoring groups based on their total scores. The top 27% of participants, with scores exceeding 47, comprised the high-scoring group, while the bottom 27%, with scores below 40, formed the low-scoring group. An independent samples t-test was performed to compare the total scores of these groups. The analysis indicated that the high-scoring group (M = 51.361) had significantly higher scores than the low-scoring group (M = 34.723), with a statistically significant difference between the two groups ($p < 0.05$).

### 4.4 Exploratory factor analysis

Exploratory Factor Analysis (EFA) was conducted on the 13 items using JASP software [89]. The adequacy of the sample was confirmed by the Kaiser-Meyer-Olkin (KMO) test, which yielded a value of 0.951. Additionally, Bartlett's test of sphericity produced a highly significant result, with $\chi2(78) = 4761.518$ ($p < 0.001$), indicating that the sample was suitable for analysis.

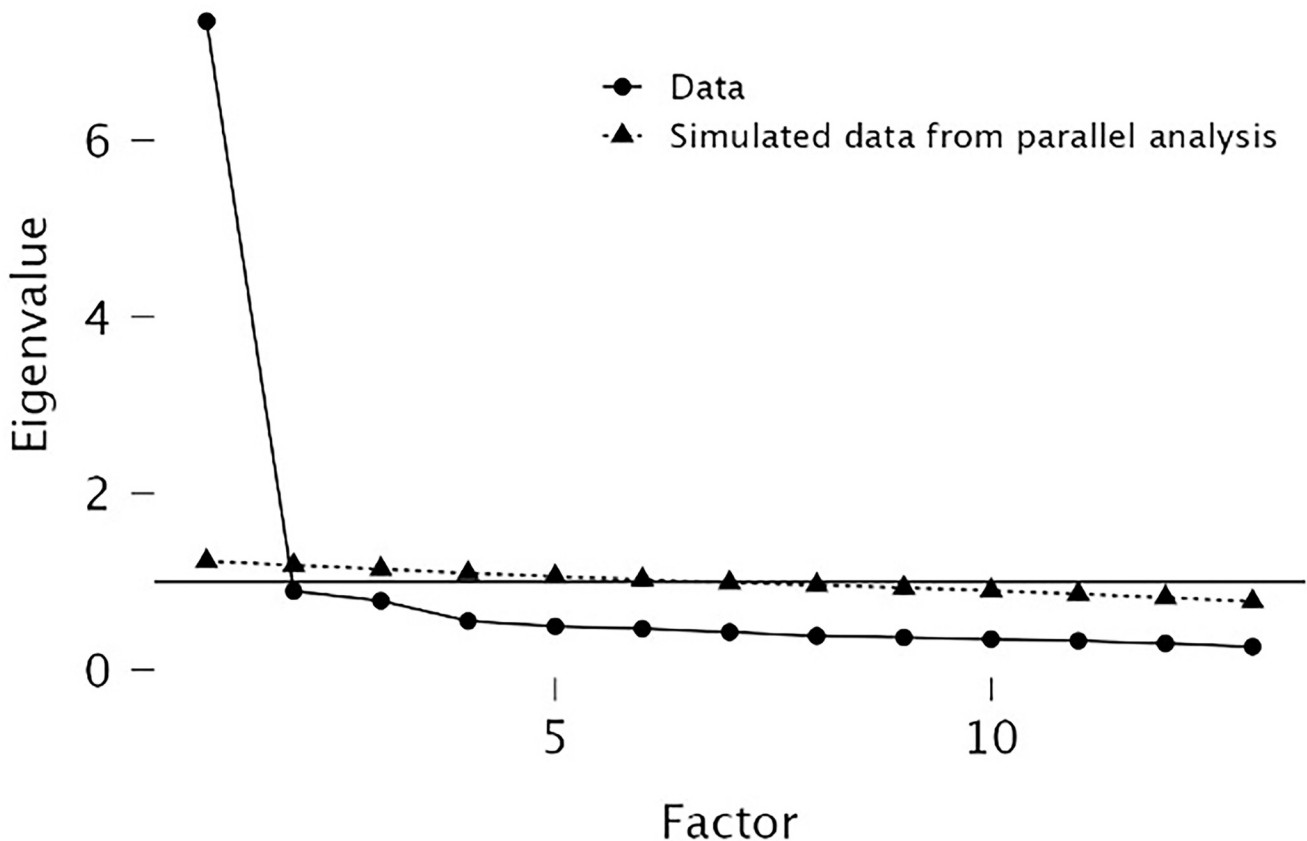

**Fig 1. Scree plot based on parallel analysis.**

Based on the findings from parallel analysis and the scree plot presented in Fig 1, three factors were identified. The first factor was designated as "Language proficiency", because it involved items that assessed the participants' perceived competence in English language skills. The second factor was named "Debating skills", because it encompassed items related to the mastery of argumentation techniques. These items captured the participants' perceived efficacy in effectively presenting and defending arguments during a debate. The third factor was labelled "Team collaboration", because it comprised items that reflected the participants' confidence in their ability to work cooperatively with teammates and contribute effectively to team dynamics. This factor highlighted the importance of teamwork and collaboration in English debate, acknowledging the role of cooperation within a debating team.

As shown in Table 2, the first factor, Language proficiency (Cronbach's Alpha = .894) explained 25.6% of the total variance, the second factor, Debating skills (Cronbach's Alpha = .861) explained 15.9% of the total variance and the third factor, Team collaboration (Cronbach's Alpha = .831), accounted for 19.2% of the total variance.

## 4.5 Construct validity

A Confirmatory Factor Analysis (CFA) was conducted to validate the three-factor structure of the EDSS, as illustrated in Fig 2. The results showed that all items were retained, with no need for elimination. The fit indices obtained were $\chi2 = 184.404$, df = 62, $\chi2/df = 2.974$, CFI = .975,

Table 2. Results of EFA of the 13-item EDSS.

| Dimension | Item | Factor 1 | Factor 2 | Factor 3 | Uniqueness | Communality |
|---|---|---|---|---|---|---|
| Debating skills | DS1 | | 0.787 | | 0.362 | 0.638 |
| | DS2 | | 0.770 | | 0.362 | 0.638 |
| | DS3 | | 0.582 | | 0.463 | 0.537 |
| | DS4 | | 0.719 | | 0.369 | 0.631 |
| Team collaboration | TC1 | | | 0.626 | 0.443 | 0.557 |
| | TC2 | | | 0.891 | 0.283 | 0.717 |
| | TC3 | | | 0.688 | 0.397 | 0.603 |
| Language proficiency | LP1 | 0.567 | | | 0.391 | 0.609 |
| | LP2 | 0.701 | | | 0.394 | 0.606 |
| | LP3 | 0.502 | | | 0.471 | 0.529 |
| | LP4 | 0.745 | | | 0.422 | 0.578 |
| | LP5 | 0.719 | | | 0.434 | 0.566 |
| | LP6 | 0.882 | | | 0.316 | 0.684 |

Note. The applied rotation method is promax.

TLI = .969, SRMR = .027, and RMSEA = .055 (90% CI, .046-.065), indicating a strong fit between the proposed three-factor model and the observed data.

## 4.6 Convergent validity

After confirming the factor structure through CFA, we proceeded to assess the convergent validity of the EDSS to provide additional evidence for its construct validity [90]. Convergent validity was evaluated by calculating the standardized factor loadings, composite reliability (CR), and average variance extracted (AVE), with acceptable thresholds set at greater than 0.50, 0.70, and 0.50, respectively [91]. As shown in Fig 2, all items displayed high loadings on their corresponding constructs, ranging from 0.70 to 0.80, significantly exceeding the recommended threshold of 0.50. The CR values for the three factors, detailed in Table 3, varied from 0.803 to 0.905, all above the minimum requirement of 0.70. Moreover, the AVE values for the three factors were 0.564, 0.577, and 0.613, each exceeding the 0.50 benchmark and remaining below their respective CR values. These findings indicate that the EDSS exhibits acceptable convergent validity.

## 4.7 Discriminant validity

Discriminant validity of the scale was assessed using the Heterotrait-Monotrait (HTMT) ratio of correlations. The analysis as shown in Table 4 revealed that all HTMT values for the scale were below the 0.90 threshold [92], indicating that the criteria for strong discriminant validity were satisfied.

## 4.8 Concurrent validity

In the current study, the concurrent validity of the EDSS was evaluated by the Pearson correlation coefficients with the English Public Speaking Self-Efficacy Scale (EPSSS) [82]. The results presented in Table 5 show that the three factors, along with the average score of the EDSS, were significantly and positively correlated with the English Public Speaking Scale. These correlations provide compelling evidence for the concurrent validity of the three-factor structure of the EDSS.

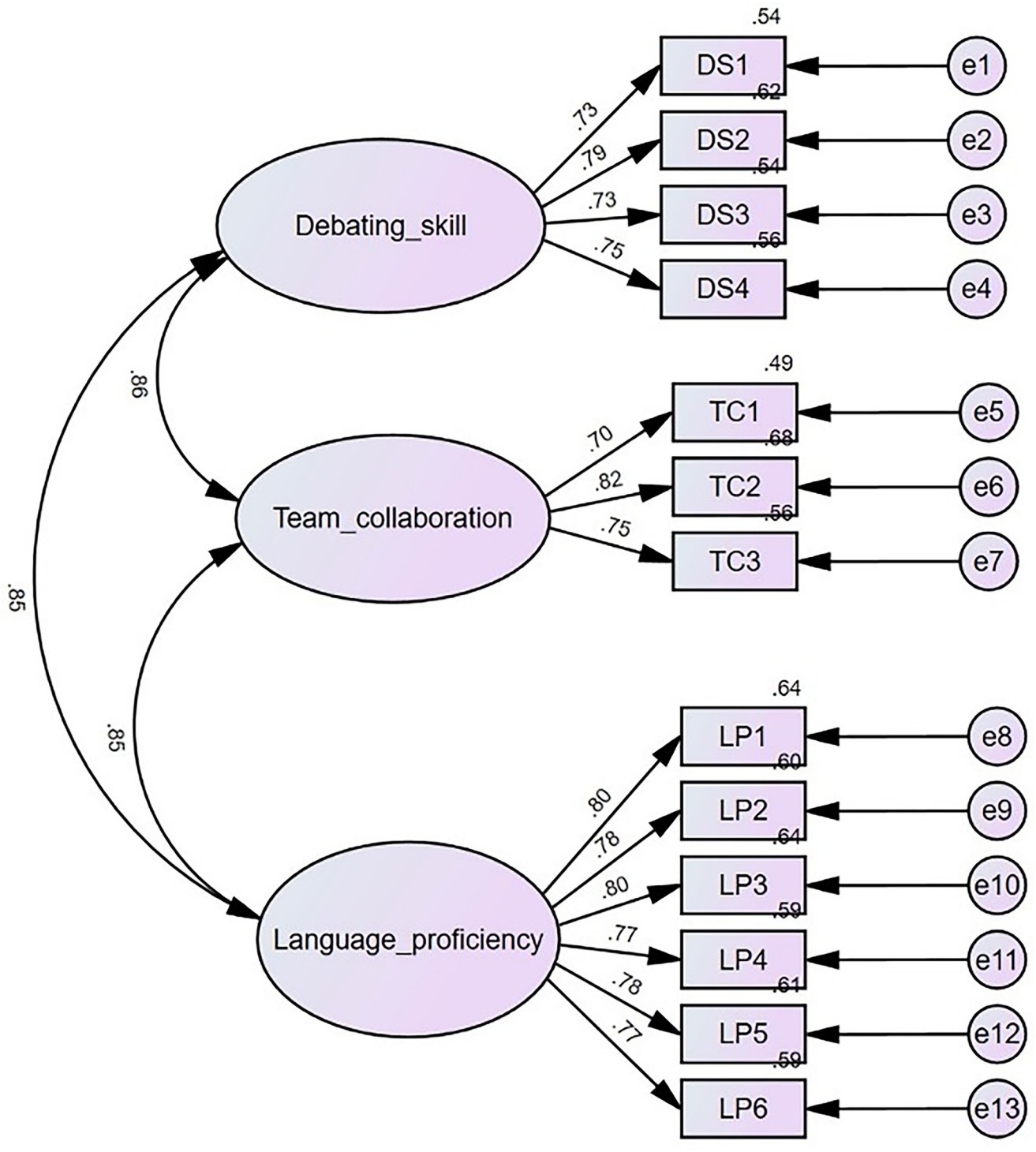

**Fig 2. Results of the CFA of the three-factor model of EDSS.**

**Table 3. Convergent validity of EDSS.**

| Factor | CR | AVE |
|---|---|---|
| Debating skill | 0.838 | 0.564 |
| Team collaboration | 0.803 | 0.577 |
| Language proficiency | 0.905 | 0.613 |

Abbreviations: CR = Composite Reliability, AVE = Average Variance Extracted

**Table 4. Heterotrait-Monotrait results.**

| Factor | Debating skill | Team collaboration | Language proficiency |
|---|---|---|---|
| Debating skill | | | |
| Team collaboration | 0.862 | | |
| Language proficiency | 0.851 | 0.855 | |

Note: HTMT = Heterotrait-Monotrait Ratio of Correlations

**Table 5. Correlation coefficients between the EDSS and the EPSSS.**

| | DS | TC | LP | EDS | EPSSS |
|---|---|---|---|---|---|
| DS | – | | | | |
| TC | .706** | – | | | |
| LP | .741** | .728** | – | | |
| EDS | .899** | .899** | .913** | – | |
| EPSSS | .664** | .539** | .682** | .695** | – |

Note: ** Correlation is significant at the 0.01 level (2-tailed).

Abbreviations: DS = Debating Skill, TC = Team Collaboration, LP = Language Proficiency, EDSS = English Debate Self-efficacy Scale, and EPSSS = English Public Speaking Self-Efficacy Scale

## 5. Discussion

The purpose of this article was to describe the development of a new questionnaire to measure Chinese college students' perceived self-efficacy belief with regard to English debate. The primary rationale for constructing a new instrument was the nature of the domain specificity of self-efficacy and the lack of a proper tool to access students' perceived self-efficacy belief with regard to English debate.

To establish the content validity of the questionnaire, we interviewed four experienced students in English debate competitions. Using MAXQDA software, we analysed the interview transcripts and identified three themes: debating skills, collaboration, and English language proficiency. We sought feedback from two skilled teachers who coached debate teams and removed three items based on their input. This rigorous process ensures the questionnaire's content validity to assess perceived self-efficacy beliefs with regard to English debate.

Based on the results of EFA, three factors were identified. The items in the first factor, Language Proficiency, reflected the participant's ability to understand and comprehend the viewpoints expressed by both their team members and opponents in English (Item LP1), highlighted the participant's capability to accurately express their viewpoints in English (Item LP2), emphasized the ability to utilize appropriate and diverse English vocabulary to enhance the persuasiveness of the debate (Item LP3), highlighted their ability to use specific English intonation to convey specific viewpoints, attitudes, or emotions (Item LP4) as well as the

competence to use appropriate pauses to prevent ambiguity (Item LP5), and stressed their competence to adapt their intonation and pace in English to different contexts and audiences (Item LP6). The items in the second factor, Debating Skill, reflected the participant's ability to effectively construct clear and persuasive arguments (Item DS1), capability to provide compelling evidence to support their position (Item DS2), competence to identify logical errors between the opponent's arguments and evidence during a debate (Item DS3), and ability to adapt and modify debate strategies in response to opponents (Item DS4). The items in the third factor, Team Collaboration, highlighted the participant's active involvement in team discussions, willingness to share their viewpoints, and their ability to listen to and respect the opinions of other team members (Item TC1), also focuses on the participant's ability to promptly recognize and address errors that may arise from the neglect of team members (Item TC2), and emphasized close collaboration with debate team members to ensure that their arguments complement and reinforce each other, leading to a strong overall performance (Item TC3).

Rigorous validation procedures of CFA with a different sample corroborated the structure of English Debating Self-efficacy as evidenced in the validity evidence with regard to structural validity, convergence validity and discriminant validity. In addition, significant correlations of the English Debate Self-Efficacy Scale (EDSS) with the English Proficiency Self-Efficacy Scale (EPSS) provided evidence of the criterion validity of the scale. These correlations are consistent with previous literature [93], which indicates that debate is closely linked to public speaking. This relationship can be explained by the fact both require effective language use and the ability to articulate thoughts clearly in front of an audience, suggesting that improvements in debate can positively impact public speaking. These analyses further support the factor structure of the EDSS and its utility for comparative research.

The findings of this study revealed that the EDSS has adequate psychometric quality in terms of reliability and validity. The CFA results provided empirical support for the existence of three separate yet related dimensions for the EDSS.

## 6. Implications

Developing domain-specific measures for English debate self-efficacy is vital for several interconnected reasons. Firstly, it enhances understanding of individuals' confidence and perceived competence in the specific skill of debate. By providing a tailored assessment tool, researchers can identify factors that influence self-efficacy, leading to more targeted interventions in foreign language education.

Secondly, this research contributes to the broader field of foreign language education by filling a significant gap in the literature. It enables educators to better understand the dynamics of self-efficacy within the context of debate, which can inform curriculum design and teaching methodologies.

Lastly, the insights gained from the English Debate Self-efficacy Scale (EDSS) can be invaluable for practical applications. For students, a clear measure of their self-efficacy can motivate them to engage more actively in debate activities, enhancing their learning experience. Coaches can use the findings to tailor their training strategies, helping participants build the necessary confidence and skills for effective performance. Furthermore, teachers can leverage this research to foster a supportive learning environment that encourages student participation and growth in debate skills.

## 7. Limitations and suggestions for future studies

Although this study has yielded meaningful results, there are also some research limitations that need to be acknowledged.

This study is subject to a significant limitation concerning the lack of sample heterogeneity. The recruitment of participants exclusively from one college in Hebei Province restricts the generalizability of the findings to a broader population. To overcome this limitation, future research should strive to incorporate participants from various colleges or universities, encompassing a more diverse demographic range. By doing so, the study can capture a broader spectrum of perspectives and experiences, leading to a more comprehensive understanding of the phenomenon being investigated.

The second limitation of this study pertains to its utilization of a cross-sectional design, which precludes the assessment of the scale's longitudinal measurement invariance by examining the responses of the same sample at different time points. To overcome this limitation, future studies could adopt a longitudinal research design to longitudinally track the responses of a consistent group of participants across multiple time points. Such a design would facilitate a more comprehensive understanding of the scale's temporal stability and consistency.

## 8. Conclusion

In conclusion, the development and validation of the English Debating Self-Efficacy Scale (EDSS) have yielded promising results, indicating its potential as a reliable and valid instrument for assessing students' perceived self-efficacy beliefs regarding English debate. The rigorous process of scale development and validation has led to the establishment of a three-factor structure, which encompasses Language Proficiency, Debating Skills, and Team Collaboration.

These factors reflect the multifaceted nature of self-efficacy in debate contexts, offering a comprehensive framework for evaluation. Overall, the findings underscore the suitability of the EDSS as a robust tool for assessing students' self-efficacy beliefs across various research and applied settings, including educational environments and training programs.

Moreover, the EDSS can facilitate targeted interventions aimed at enhancing students' confidence and performance in English debate, thereby contributing to their overall communication skills and development as proactive participants in global discussions. Future studies should continue to examine the scale's psychometric properties, including its reliability and validity, while also exploring its utility across diverse populations and contexts. This will ensure that the scale remains relevant and effective in measuring self-efficacy in different educational and cultural settings. Ultimately, the ongoing refinement and application of the EDSS can support educators in fostering a more engaging and empowering learning environment for students involved in English debate.

## Supporting information

**S1 Table. Appendix A: English Debating Self-Efficacy Scale.**
(DOCX)

**S1 Dataset.**
(XLSX)

**S2 Dataset.**
(XLSX)

## Author Contributions

**Conceptualization:** Yanchao Yang, Feng (Robin) Wang.

**Data curation:** Wangze Li.

**Formal analysis:** Wangze Li.

**Investigation:** Fanghua Liu, Yanchao Yang.

**Methodology:** Fanghua Liu, Yanchao Yang.

**Project administration:** Fanghua Liu.

**Validation:** Yanchao Yang, Feng (Robin) Wang.

**Visualization:** Yanchao Yang.

**Writing – original draft:** Yanchao Yang.

**Writing – review & editing:** Fanghua Liu.

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
