## [Decision Letter · Decision Letter 0]

24 Oct 2024

PONE-D-24-33242The English Debating Self-efficacy Scale: Scale Development, Validation, and Psychometric PropertiesPLOS ONE

Dear Dr. Yang,

Thank you for submitting your manuscript to PLOS ONE. After careful consideration, we feel that it has merit but does not fully meet PLOS ONE’s publication criteria as it currently stands. Therefore, we invite you to submit a revised version of the manuscript that addresses the points raised during the review process.

The reviewers have provided constructive feedback to improve the quality of the paper, “The English Debating Self-efficacy Scale: Scale Development, Validation, and Psychometric Properties.” The abstract needs restructuring to clearly state the study's gap, key findings, and conclusion. In the introduction, the research gap, problem statement, objectives, and purpose should be more clearly articulated. The literature review requires subheadings for better organization and flow. Significant revisions are needed in the methodology section, including clear statements on research design, population, item generation, and validation processes. Ethical considerations should be placed at the end of the methodology. The instrument section should clarify how the items were generated, the number of items included and eliminated, and the criteria for elimination. The discussion section requires more citations to support the findings. Expanding the scale dimensions, employing a longitudinal design, and including test-retest reliability will strengthen the study. The implications section should elaborate on the EDSS’s practical applications across various settings. 

Kind regards,

Musa Adekunle Ayanwale

Academic Editor

PLOS ONE

We look forward to receiving your revised manuscript.

Kind regards,

Musa Adekunle Ayanwale

Academic Editor

PLOS ONE

https://pubmed.ncbi.nlm.nih.gov/38789483/

In your revision ensure you cite all your sources (including your own works), and quote or rephrase any duplicated text outside the methods section. Further consideration is dependent on these concerns being addressed.

3. We are unable to open your Supporting Information file [EFA.sav and CFA.sav]. Please kindly revise as necessary and re-upload.

Reviewers' comments:

Reviewer's Responses to Questions

**Comments to the Author**

1. Is the manuscript technically sound, and do the data support the conclusions?

Reviewer #1: Partly

Reviewer #2: Partly

2. Has the statistical analysis been performed appropriately and rigorously? 

Reviewer #1: Yes

Reviewer #2: Yes

3. Have the authors made all data underlying the findings in their manuscript fully available?

Reviewer #1: Yes

Reviewer #2: Yes

4. Is the manuscript presented in an intelligible fashion and written in standard English?

Reviewer #1: No

Reviewer #2: Yes

5. Review Comments to the Author

Reviewer #1: Dear Author, several areas require improvement to enhance the research's quality and impact. The abstract needs restructuring to provide a clearer overview of the study's key elements. The introduction should be reorganized to clearly state the research gap, problem statement, objectives, and purpose. The literature review lacks structure and requires subheadings for better organization. The methodology section needs significant revision, including a clear statement of the research design and methods employed. The discussion section requires more citations to support or argue the findings.

To strengthen the study, consider expanding the scale's dimensions, employing a longitudinal design, and including test-retest reliability. The implications section should elaborate on the EDSS's practical applications in various settings. These improvements will enhance the scale's validity, reliability, and usefulness in assessing and developing English debate skills.

Reviewer #2: It is a great pleasure to have reviewed this paper, “The English Debating Self-efficacy Scale: Scale Development, Validation, and Psychometric Properties”. The study has numerous merits, but for this review, I will unravel some grey areas that, after implementing the correction, will further strengthen the quality of the paper.

Abstract

In a sentence, the authors should state the gap in the study. The gap in the study lies in the lack of detailed analysis of the variables that may have influenced the result of absence. Results were absent in the abstract, and the conclusion was not well crafted. Additionally, the conclusion fails to summarize the key findings and implications of the study effectively.

Methodology

The study fails to identify its design as well as the selection process.

How were the items generated?

Who are the study population?

Prior to the distribution of the question, were the items validated? If yes, how was it validated? What type of validation was conducted?

Ethical Consideration

I think the ethical considerations should be the late subsection after the analytical procedure in the methodology.

Instrument

How were the items generated? Before the analysis, how many items were included in the questionnaire? How many items were eliminated? What were the criteria for such elimination? Which item(s) was/were eliminated, if any?

References

Some of the references are not complete. Ensure to include the doi for those that have it.

6. PLOS authors have the option to publish the peer review history of their article (what does this mean?). If published, this will include your full peer review and any attached files.

Reviewer #1: **Yes: **Damola Olugbade

Reviewer #2: **Yes: **Oluwaseyi Aina Gbolade Opesemowo

---

## [Author Response · Author response to Decision Letter 0]

5 Nov 2024

Due to the extensive nature of our replies, which exceed the 2,000-word limit, we have compiled our responses to the comments in a Word document for your convenience. Please find it attached for your review.

---

## [Decision Letter · Decision Letter 1]

19 Nov 2024

The English Debating Self-efficacy Scale: Scale Development, Validation, and Psychometric Properties

PONE-D-24-33242R1

Dear Dr. Yang,

We’re pleased to inform you that your manuscript has been judged scientifically suitable for publication and will be formally accepted for publication once it meets all outstanding technical requirements.

Kind regards,

Musa Adekunle Ayanwale

Academic Editor

PLOS ONE

Additional Editor Comments (optional):

Reviewers' comments:

Reviewer's Responses to Questions

**Comments to the Author**

1. If the authors have adequately addressed your comments raised in a previous round of review and you feel that this manuscript is now acceptable for publication, you may indicate that here to bypass the “Comments to the Author” section, enter your conflict of interest statement in the “Confidential to Editor” section, and submit your "Accept" recommendation.

Reviewer #1: All comments have been addressed

Reviewer #2: All comments have been addressed

2. Is the manuscript technically sound, and do the data support the conclusions?

Reviewer #1: Yes

Reviewer #2: Yes

3. Has the statistical analysis been performed appropriately and rigorously? 

Reviewer #1: Yes

Reviewer #2: Yes

4. Have the authors made all data underlying the findings in their manuscript fully available?

Reviewer #1: Yes

Reviewer #2: Yes

5. Is the manuscript presented in an intelligible fashion and written in standard English?

Reviewer #1: Yes

Reviewer #2: Yes

6. Review Comments to the Author

Reviewer #1: (No Response)

Reviewer #2: The authors worked diligently to resolve all issues that were brought up during the review. They contributed their ideas and expertise to come up with effective solutions. After thorough evaluation, the authors left the reviewer feeling confident and motivated to consider the article for publication.

7. PLOS authors have the option to publish the peer review history of their article (what does this mean?). If published, this will include your full peer review and any attached files.

Reviewer #1: **Yes: **Damola Olugbade

Reviewer #2: **Yes: **Opesemowo Oluwaseyi Aina Gbolade

---

## [Editor Report · Acceptance letter]

21 Nov 2024

PONE-D-24-33242R1 

PLOS ONE

Dear Dr. Yang, 

I'm pleased to inform you that your manuscript has been deemed suitable for publication in PLOS ONE. Congratulations! Your manuscript is now being handed over to our production team.

Kind regards, 

on behalf of

Dr. Musa Adekunle Ayanwale 

Academic Editor

PLOS ONE